# Validating METCAM/MUC18 as a Novel Biomarker to Predict the Malignant Potential of Prostate Cancer at an Early Stage by Using a Modified Gold Nanoparticles-Based Lateral Flow Immunoassay

**DOI:** 10.3390/diagnostics11030443

**Published:** 2021-03-04

**Authors:** Jui-Chuang Wu, Yin-Huan Chuang, Yu-Chun Wei, Chia-Chi Hsieh, Yuan-Hung Pong, Yenn-Rong Su, Vincent F.-S. Tsai, Guang-Jer Wu

**Affiliations:** 1Biochemical Engineering Laboratory, Department of Chemical Engineering, Chung Yuan Christian University, Chung-li District, Taoyuan City 32023, Taiwan; ph1015p@gmail.com (Y.-H.C.); yuchunwei1022@gmail.com (Y.-C.W.); lena100019@gmail.com (C.-C.H.); 2Research Center for Circular Economy, Chung Yuan Christian University, Chung-Li District, Taoyuan City 32023, Taiwan; 3Department of Urology, Ten Chen General Hospital, Yang-Mei District, Taoyuan City 326, Taiwan; pong9101095@yahoo.com.tw (Y.-H.P.); 24162@office.cycu.edu.tw (V.F.-S.T.); 4Department of Urology, Ten Chan General Hospital, Chung-li District, Taoyuan City 320, Taiwan; 5Department of Urology, National Taiwan University Hospital Hsin-Chu Branch, Hsin-Chu City 300, Taiwan; 105888001@cc.ncu.edu.tw; 6Cancer Metastasis Laboratory, Department of Bioscience Technology, Chung Yuan Christian University, Chung-li District, Taoyuan City 32023, Taiwan; 7Molecular Biology of Cancer Metastasis Laboratory, Department of Microbiology and Immunology, Emory University School of Medicine, Atlanta, GA 30322, USA

**Keywords:** biomarker, biosensor, METCAM/MUC18, human serum, modified LFIA, biotinylated antibodies, nano-gold conjugated antibodies, streptavidin, malignant potential, prostate cancer

## Abstract

(1) Background: To further validate METCAM/MUC18 as a diagnostic biomarker for prostate cancer, a modified Lateral Flow Immune Assay (LFIA) with increased sensitivity and specificity was designed by taking advantage of the extremely high affinity between biotin and streptavidin and used. (2) Methods: The combination of a commercial biotinylated rabbit antibody (EPP11278), or the home-made biotinylated chicken antibody, and the nano-gold conjugated home-made chicken antibody or a commercial rabbit antibody (EPP11278), had the higher sensitivity and specificity in this modified LFIA to establish calibration curves from the two recombinant METCAM/MUC18 proteins and were used for determining METCAM/MUC18 concentrations in serum specimens from normal individuals, benign prostatic hyperplasia (BPH) patients, prostatic intraepithelial neoplasia (PIN) patients, prostate cancer patients with various Gleason scores, and treated patients. (3) Results: Data obtained by this modified LFIA were statistically better than traditional LFIA and prostate-specific antigen (PSA) test. Interestingly, serum METCAM/MUC18 concentrations were higher in pre-malignant PIN patients than prostate cancer patients and both were higher than normal individuals, BPH patients, and treated patients. Serum METCAM/MUC18 concentrations were directly proportional to most serum PSA. (4) Conclusions: Elevated serum METCAM/MUC18 concentrations may be used for predicting the malignant potential of prostate cancer at an early premalignant (PIN) stage, which is not achievable by the current PSA test.

## 1. Introduction

Prostate cancer is the top-leading cause of cancer in males and one of the major causes of cancer death in the world [1]. Nevertheless, majority of prostate cancer is indolent; however, about 1/10 of prostate cancer are aggressive, from which the patients may succumb to death within one year of the first diagnosis [2,3]. The most common diagnosis test for the cancer is to detect an elevated serum prostate-specific antigen (PSA) level [4]. However, the elevated level of PSA in the serum is not always prognostic of the correct pathologic stage of the cancer or the presence of an indolent or a metastatic disease; thus, it results in at least 20–25% of false diagnosis [4]. Many potential biomarkers for predicting prostate cancer progression have been published [5,6,7,8] and some of them are even Gleason grade-specific biomarkers [8]. However, only PSA, Prostate Health Index, and PCA3 have been approved by Food and Drug Administration for the diagnosis of prostate cancer [5]. Therefore, to find a better diagnostic marker for accurate predicting the malignant propensity of prostate cancers is still an urgent need. The new diagnostic marker should at least compensate for, or even better to replace the present serum PSA test. We have previously showed that METCAM/MUC18, an immunoglobulin-like cell adhesion molecule [9], is capable of driving the spread of prostate cancer to multiple organs [9,10,11,12,13,14,15,16,17]. Furthermore, based on our immunohistochemistry results, METCAM/MUC18 was overly expressed in most (80%) prostate cancer at advanced pathological stages and in all patients (100%) with the premalignant PIN (prostatic intraepithelial neoplasia); however, it was not expressed in most (80%) normal prostates nor in the prostates of all (100%) the benign prostatic hyperplasia (BPH) patients [9,10,11]; we suggest that it has the high potential to be used as a novel diagnostic biomarker for predicting the metastatic potential of human prostate cancer at an early pre-malignant stage (PIN) [9,10,11,16], and perhaps for distinguishing the aggressive cancers from indolent ones [15,16,17].

To test the above hypothesis, we first showed to be able to use three immunological methods, such as the immunoblot (Western blot) analysis, the enzyme-linked immunosorbent assay (ELISA), and membrane-based lateral flow immunoassay (LFIA), to detect the possible presence of the METCAM/MUC18 antigen in human serum specimens [18]. Then, we further demonstrated that the METCAM/MUC18 concentrations were elevated in the serum of the prostate cancer patients, suggesting that the elevated serum METCAM/MUC18 concentrations may be used for predicting the malignant potential of prostate cancer and at least complement the PSA test [18]. However, the first two methods were consisted of many labor-intensive steps, which may create errors in human operators, resulting in irreproducible data. Moreover, the serum concentration of METCAM/MUC18 was directly proportional to that of PSA only when PSA concentrations were less than 7 ng/mL, but not when PSA was between >7 ng/mL and 1219 ng/mL. Though the third method was better than the two methods in sensitivity and also in the aspect of being proportional to PSA, it somewhat also had limited sensitivity. This could be due to the interfering substances in human serum when polyclonal antibodies were used for the assay as pointed out by previously published findings [19].

To overcome these obstacles and to further validate METCAM/MUC18 as a diagnosis marker for the cancer, the purpose of this research was to improve and modify the traditional LFIA test to increase the sensitivity and specificity of the assay by taking advantage of the extremely high affinity between biotin and streptavidin [20]. In brief, in this modified LFIA the biotinylated primary antibodies were first allowed to interact with the antigen and the nano-gold conjugated another primary antibody, which recognized different epitopes, to form ternary complexes. The ternary complexes were then loaded on the conjugate pad in the LFIA assembly and further developed into the nitrocellulose (NC) membrane. The streptavidin was sprayed on the test line of the NC membrane to facilitate capturing the complexes. The secondary goat Abs against the primary chicken or rabbit antibody were sprayed at the control line to capture the excess unbound nano-gold conjugated primary Abs [21,22,23].

In this report, we used two different combinations of biotinylated chicken or rabbit antibodies and nano-gold-conjugated chicken or rabbit antibodies to hopefully obtain the data with the least standard deviations and with the best statistically significant results. We found that both the combinations of a commercial biotinylated rabbit anti-huMETCAM/MCU18 (EPP11278) and nano-gold-conjugated home-made chicken antibody and that of a biotinylated home-made chicken anti-huMETCAM/MUC18 Ab with a commercial rabbit antibody (MBS2529469 or EPP11278) fulfilled the expectation. Different concentrations of the recombinant METCAM/MUC18 proteins, NM-GST, and C-terminus-GST, were used as the positive and the negative controls, respectively. Under optimal conditions, the signals of the positive control protein, NM-GST, and the negative control protein, C-terminus-GST, could be clearly differentiated with a significant 5 to 8-fold difference; thus, a standard calibration curve of human recombinant antigens was established. By comparing the signals from various human serum samples to the standard calibration curve, the concentrations of METCAM/MUC18 in the serum samples could be determined. Thus, this modified LFIA method not only had a higher sensitivity and specificity to differentiate the positive and negative control proteins, but also was able to determine the huMETCAM/MUC18 protein concentration in human serum specimens. It also showed that the serum METCAM/MUC18 concentrations were proportional to most serum PSA concentrations. The standard deviations of the METCAM/MUC18 concentrations determined by this modified LFIA were significantly smaller and, therefore, better than the traditional LFIA and the PSA test. Similar to our previous findings, the serum METCAM/MUC18 concentrations from the prostate cancer patients were significantly higher than normal individuals, BPH patients, and the treated patients [18,21,22,23]. The most significant finding was that the serum METCAM/MUC18 concentrations from the pre-malignant PIN patients were higher than the prostate cancer patients [21,22,23]. Based on the evidence presented in this report, this modified LFIA had a high potential to become an accurate, simple, and rapid diagnostic test for accurate prediction of the malignant propensity of clinical prostate cancer at the premalignant stage, which is different from the PSA test.

## 2. Materials and Methods

### 2.1. Materials

Human serum samples were from 8 normal individuals, 4 patients with BPH, 2 with PIN, 14 with high-grade prostate cancer, and 2 treated patients. Chicken anti-human METCAM/MUC18 was home-made [9]. Chicken anti-murine METCAM/MUC18 was also home-made [24]. The rabbit anti-METCAM/MUC18 antibody, which recognized the N-terminal epitopes aa#26-350 of METCAM/MUC18, with or without biotin-conjugation was from Elabscience (EPP11278) (Houston, TX, USA) or from MyBiosource (MBS2529469) (San Diego, CA, USA). Goat anti-rabbit antibody (2 mg/mL) and goat anti-chicken antibody (1 mg/mL) were from Jackson lab Inc (Bar Harbor, ME, USA). BSA (Cat #AD0023), sucrose (Cat # DB0194), and Tween 20 (Cat #TB0560) were from Bio Basic Inc. (Toronto, Canada). BSA and BSA-FV were also from Promega (Madison, WI, USA). Tween 20 was also from Promega. Triton X-100 (Cat # DB0198) was from Sigma-Aldrich Co (St. Louis, MO, USA). Nitrocellulose membrane (Cat # M10-00101), sample pad (Cat # M10-01401), conjugated pad (Cat # M10-00701), absorbent pad (Cat # M10-01701), and plastic backing pad (Cat # M10-00400) were from Rega Biotech Inc., (Taipei, Taiwan). Pall Vivid 170 nitrocellulose membrane (Cat # VIV1702503R) from Pall Co, USA (Port Washington, NY, USA) (Rainbow Biotech Co., Taipei, Taiwan) also was used for LFIA. Sample pad, conjugate pad, and plastic backing pad (Cat # ARcare^®^ 9021D) were also from Prisma Biotech Co (Taipei, Taiwan). Absorption pad (Cellulose Fiber Sample Pads (CFSP203000)) was also from BERTEC Biotechnology (Torrance, CA, USA). Colloidal gold (40 nm) was from Rega Biotechnology. Streptavidin (1 mg/mL) was from Rainbow Biotechnology (Taipei, Taiwan). General chemicals were from Bio Basic (Toronto, ON, Canada).

### 2.2. Preparation of Recombinant METCAM/MUC18 Proteins

The recombinant METCA/MUC18 proteins were obtained and purified as described [9,24]. The final protein concentration of each recombinant protein was determined by measuring the A_280_ in a spectrophotometer or quantitated by a Biuret method (BioRad), verified by SDS-PAGE and staining. The recombinant proteins were kept frozen at −20 °C till use. Two recombinant proteins, NM-GST, C-terminus-GST were used for establishing standard calibration curves.

### 2.3. Preparation of Human Serum Samples

Human whole bloods were obtained from voluntary donors with the consensus and an approval of Institutional Review Board at Chung Yuan Christian University (protocol code 201706214-A1, 2017–2018, and 20180519-8, 2018–2019). Human blood (15 or 30 mL per person) was withdrawn into one or two 16 mL conical centrifuge tubes without any heparin and processed to serum by centrifugation twice at 180× *g* for 20 min in a S4180 swing out rotor in a table top centrifuge as described [18]. The final clear yellowish serum was aliquoted in 0.5 or 1mL portions in several new 1.5 mL Eppendorf centrifuge tubes and stored in −20 °C freezer until use.

### 2.4. Preparation of Biotinylated Chicken Antibody

The home-made chicken antibody was biotinylated by the using the biotin (Type A) Fast Conjugation Kit from Abcam (ab201795-300ug) [21,22,23]. The epitope-recognition of the antibody was not affected [Manual for antibody conjugation kit from Abcam]. In brief, 100 μL (1–2.5 mg/mL) of dialyzed chicken antibody was added 10 μL of modifier to form the antibody-modifier mixture. The antibody-modifier mixture was added to the vial of the lyophilized conjugation, after thoroughly mixing the vial was capped, and the reaction was allowed to occur at room temperature (26 °C) for 15 min. After addition of 10 μL quencher, the reaction was allowed to occur for 4 min and added 400 μL of PBS, and dialyzed against PBS with three changes of PBS at 4 °C. The final biotinylated antibody solution was added Na. Azide to 0.02% and stored at 4 °C till use. The final concentration was between 0.34–0.44 mg/mL. The biotinylated chicken antibody was diluted with PBS to about 0.02–0.04 mg/mL before use in LFIA. The avidity and specificity of the biotinylated antibody was verified by using Western blot analyses to test the recognition of their specific epitopes in whole cell lysates, which were prepared from SK-MEL 28, PC-3, DU145, or TSU-PR1 cell lines, and in various recombinant METCAM/MUC18 proteins.

### 2.5. Western Blot Analysis

The Western blot analysis was done as described [25] with slight modifications [9,10,11,12,14,15,18]. In brief, after electrophoresis the gel was electro-blotted to a nitrocellulose membrane in a BioRad mini-blot apparatus, the METCAM/MUC18 band was identified by reacting with 1/300 dilution of our home-made chicken anti-human METCAM/MUC18 recombinant protein internal epitopes aa# 212-374 IgY [9], subsequently with 1/2000 dilution of an AP-conjugated rabbit anti-chicken antibody, and finally with a color reaction as described [9,10,11,12,14,15,18]. The METCAM/MUC18 band on the nitrocellulose membrane was imaged by an Epson Perfection V330 Photo scanner and stored as a JPG file and quantified by the Image J software.

### 2.6. The Modified Lateral Flow Immunoassay (LFIA)

LFIA was carried on a 0.4 cm × 2.5 cm nitrocellulose membrane strip as described [18,21,22,23] and the set-up of LFIA is illustrated in the following Figure 1.

In brief, the test line (T-line) contained 0.53 μg of streptavidin per 0.4 cm strip. The control line (C-line) contained either 0.53 μg of goat antibody against primary chicken antibody or 1.07 μg of goat antibody against primary rabbit antibody per 0.4 cm strip. Both were printed with a AgitestTM RP-100 printing machine (Rega Biotech Inc., Taiwan) on a 30 × 25 cm nitrocellulose membrane (Cat # M10-00101, Rega Biotech co), which was baked for 30 min, cooled, and kept in a dehumidifier, and cut into about 75 of 0.4 cm × 2.5 cm strips with a computer-operated automatic micro cutter machine (model JS-101, HSHUENN, Taiwan) and used for the assembly of LFIA. The lateral flow buffer contained 1XPBS, 10 mM Tris HCl, pH 7.6, 0.15 M NaCl, 10 mM MgCl2, 1 mM ZnCl2, 0.1% Triton X-100, 0.5% (5 mg/mL) BSA or BSA-FV, and 0.02% Na. Azide. Conjugation of a primary antibody (either chicken or rabbit antibody) to colloidal gold (40 nm particles) was carried out at 0 °C or room temperature for 2 h, centrifuged, washed with cold 10 mM Tris.HCl buffer, pH 7.6, and finally suspended in 0.02 mL of LF buffer before use. The absorption peak of the gold nanoparticles had a slight shift from 525 nm to 530 nm, indicating that there was no aggregation after the antibody conjugation. Then, 10 μL of antibody (about 0.2 μg of rabbit antibody or 0.5 μg of chicken antibody)-gold nanoparticles (about 7.85 μg) suspension was mixed with the antigen (10 μL) and biotinylated antibody (10 μL of 47 μg/mL of biotinylated rabbit antibody or 20–40 μg/mL of biotinylated chicken antibody) and incubated at room temp for 30 min. After application of the mixture of triple complexes, the LFIA assembly was developed by 3–4 applications of 40–60 μL of LF buffer, disassembled, and only the nitrocellulose membrane strip was allowed to develop color and dried at room temperature. The images in the test line (T-line) and the control line (C-line) were scanned with an Epson Scanner and the intensities of the bands were determined by the Image J version 3.1. Each LFIA analysis of the control proteins was repeated 10 time and that of serum 7 times.

### 2.7. Statistical Analysis

Standard deviation of each set of data and R^2^ of two different sets of data were analyzed by the build-in programs in the Excel. The one-way ANOVA method in the SPSS software (IBM SPSS statistics, version 20) was used for statistical analysis among several sets of data. The difference among different sets of data was considered statistically significant if *p* value was <0.05.

## 3. Results

### 3.1. Identification of Antibodies for Modified LFIA

To identify possible the antibodies with the best specificity and sensitivity, Western blot assay method was used to analyze our home-made chicken antibody and two polyclonal rabbit antibodies from commercial sources for their abilities to recognize the full size huMETCAM/MUC18 protein in the whole cell lysates from various prostate cancer cell lines and the respective recombinant proteins that contain different epitopes. The results of the epitope recognition of these antibodies are summarized in Table 1. Our home-made chicken antibody, which recognized the amino acid residues from aa #212-374 had the best specificity and sensitivity for huMETCAM/MUC18, the next best was a commercial rabbit antibody: EPP11278, which recognized the epitopes aa #26-350. These antibodies were used for the modified LFIA, as shown in Figure 1. This report includes the results of the modified LFIA by using the two antibody combinations: (a) a commercial biotinylated rabbit antibody (EPP11278) with a nano-gold-conjugated home-made chicken antibody and (b) the home-made biotinylated chicken antibody with a nano-gold-conjugated commercial rabbit antibody (EPP11278).

### 3.2. The First Antibody Combination: A Commercial Biotinylated Rabbit Antibody (EPP11278) and the Nano-Gold Conjugated Home-Made Chicken Antibody

#### 3.2.1. The Standard Curve of Recombinant Human METCAM/MUC18 Proteins

Figure 2A shows the results of color image intensities in the test lines of the modified LFIA by using the first antibody combination to determine the intensities versus concentrations of the two recombinant METCAM/MUC18 proteins, NM-GST (the positive control) and C-terminus-GST (the negative control). Figure 2B shows the quantitative results of Figure 2A after being scanned and analyzed by the Image J software. As shown in Figure 2B, the positive control (NM-GST) and the negative control (C-terminus-GST) could be clearly differentiated with a significant 3 to 8-fold difference (*p* = 0.005). After correcting the values of the positive control by subtracting the values of the negative control, a standard calibration curve was deduced and Figure 2C shows the resulting linear relationship of intensities versus concentrations (R^2^ = 0.9809). The standard calibration curve in Figure 2C was used for deducing the METCAM/MUC18 concentrations in various human serum specimens, as shown below:

#### 3.2.2. The Serum METCAM/MUC18 Concentrations in Various Human Serum Samples

Figure 3A shows the color images in the test lines of 28 serum samples from normal individuals, BPH patients, premalignant prostate cancer patients, prostate cancer patients with different Gleason score, and the treated prostate cancer patients. Table 2 shows the characteristics of 30 patients and the pathological grades of 14 prostate cancer patients. Figure 3B shows that the quantitative results after analyzing the image intensity with the Image J software. As shown in Figure 3B,C, the average concentrations of METCAM/MUC18 in human serum from the pre-malignant PIN patients and the prostate cancer patients at various Gleason scores were statistically significantly higher than normal individuals, BPH patients, and the treated patients. Figure 3B also shows that the average serum METCAM/MUC18 concentration from the pre-malignant PIN patients was the highest, about seven times higher than the normal individuals and 2.33 times higher than the prostate cancer patients. Figure 3C also shows that the serum METCAM/MUC18 concentrations appeared to linearly increase with increasing Gleason scores.

#### 3.2.3. The Relation of Serum MECTAM/MUC18 Concentrations with PSA

Figure 4A shows that the serum METCAM/MUC18 concentrations determined by this modified LFIA were linearly proportional to most of the serum PSA concentrations (R^2^ = 0.8244), when PSA was less than 12 ng/mL. Figure 4B–D show that they were somewhat exponentially proportional to most PSA, when PSA was less than 60 ng/mL (R^2^ = 0.3754), less than 350 ng/mL (R^2^ = 0.4681), and less than 1219 ng/mL (R^2^ = 0.4027), respectively.

### 3.3. The Second Antibody Combination: The Home-Made Biotinylated Chicken Antibody and a Nano-Gold Conjugated Commercial Rabbit Antibody (EPP11278 or MBS2529469)

#### 3.3.1. The Standard Curve of Recombinant Human METCAM/MUC18 Proteins

Figure 5A shows the results of the color image intensities in the test lines of the modified LFIA by using the second antibody combination to determine the intensity versus concentrations of the two recombinant METCAM/MUC18 proteins, NM-GST (the positive control) and C-terminus-GST (the negative control). Figure 5B shows the quantitative results of the Figure 5A after being scanned and analyzed by the Image J software. As shown in Figure 5B, the positive control protein (NM-GST) and the negative control protein (C-terminus-GST) could be clearly differentiated with a significant 2 to 7-fold difference (*P* = 0.011). After correcting the values of the positive control by subtracting the values of the negative control, a standard calibration curve was deduced and Figure 5C shows the resulting linear relationship of intensities versus concentrations (R^2^ = 0.9394). The standard calibration curve in Figure 5C was used for deducing the METCAM/MUC18 concentrations in various human serum specimens, as shown below:

#### 3.3.2. The Serum METCAM/MUC18 Concentrations in Various Human Serum Samples

Figure 6A shows the color images in the test lines of 28 serum samples from normal individuals, BPH patients, premalignant prostate cancer patients, prostate cancer patients with different Gleason score, and the treated prostate cancer patients. Figure 6B shows that the quantitative results after the images were scanned and their intensities analyzed with the Image J software. As shown in Figure 6B,C, the average concentrations of METCAM/MUC18 in human serum from the pre-malignant PIN patients and the prostate cancer patients at various Gleason scores were statistically significantly higher than normal individuals, BPH patients, and the treated patients. Figure 6B also shows that the average serum METCAM/MUC18 concentration from the pre-malignant PIN patients was the highest, about 11 times higher than the normal individuals and 2.2 times higher than the prostate cancer patients. Figure 6C shows that the serum METCAM/MUC18 concentrations somewhat increased with increasing Gleason scores.

#### 3.3.3. The Relation of Serum MECTAM/MUC18 Concentrations with PSA

Figure 7A shows that the serum METCAM/MUC18 concentrations determined by this modified LFIA were linearly proportional to most of the serum PSA concentrations (R^2^ = 0.7995), when PSA was less than 12 ng/mL. Figure 7B–D show that they were exponentially proportional to most PSA, when PSA was less than 60 ng/mL (R^2^ = 0.4723), less than 350 ng/mL (R^2^ = 0.5581), and less than 1219 ng/mL (R^2^ = 0.4624), respectively.

### 3.4. The Relation of Serum PSA Concentration in Various Patients

Figure 8A shows that the serum PSA concentrations in prostate cancer patients were statistically significantly higher than those in BPH patients, normal individuals, and treated patients, but not higher than those in PIN patients. However, the serum PSA concentrations in PIN patients were not statistically higher than those in BPH patients, normal individuals, and treated patients. The serum PSA concentrations in BPH patients, as anticipated, were statistically higher than normal individuals and treated patients. Figure 8B shows that the serum PSA concentrations did not statistically significantly increased with increasing Gleason scores.

## 4. Discussion

In this report, we presented evidence to show in this modified LFIA that it was possible to improve the traditional LFIA by taking advantage of an extremely high affinity between biotin and streptavidin. Using this modified LFIA with two combinations of biotinylated primary antibodies and nano-gold conjugated antibodies, we were able to establish standard calibration curves from huMETCAM/MUC18 recombinant proteins and used these curves to determine the serum METCAM/MUC18 concentrations in normal individuals, BPH patients, patients with PIN, prostate cancer patients with various Gleason scores, and treated prostate cancer patients. Both antibody combinations showed that the serum METCAM/MUC18 concentrations in the PIN patients and prostate cancer patients were consistently higher than the normal individuals, BPH patients, and the treated prostate cancer patients. It should be noted that the visible intensities of the METCAM/MUC18 band in the serum samples by using the second antibody combination (biotinylated chicken antibody and nano-gold conjugated rabbit antibody) were stronger than those by using the first antibody combination (biotinylated rabbit antibody and nano-gold conjugated chicken antibody), which appeared to be consistent with the better specificity and sensitivity of the home-made chicken antibody for huMETCAM/MUC18 than the commercial rabbit antibody (EPP11278). It was intriguing that the serum METCAM/MUC18 concentrations appeared to increase with increasing Gleason scores, suggesting that serum METCAM/MUC18 concentrations increased with increasing pathological stages and perhaps with the increasing degree of malignancy of the cancer. Surprisingly, the serum METCAM/MUC18 concentrations in the PIN patients were consistently higher than the prostate cancer patients, which could not be achieved by our previous traditional LFIA and ELISA test [18] and also the current PSA test (Figure 8). We also showed that the serum METCAM/MUC18 concentrations were proportional to most of the PSA concentrations. Taken together, we have successfully developed a reliable, cost-effective test to prove not only the use of METCAM/MUC18 as a diagnostic biomarker for predicting the malignant potential of prostate cancer, but also better than a traditional LFIA and the PSA tests in that the elevated serum METCAM/MUC18 can be used to predict the malignant propensity of prostate cancer at the premalignant (PIN) stage before becoming a frank malignant cancer (namely a metastatic cancer), which cannot be achieved by the PSA test (Figure 8), suggesting that METCAM/MUC18 has the advantage over the PSA test in early detection of the malignant potential of prostate cancer.

The major advantages of this modified LFIA are that the procedures are relatively simple, user friendly and easy operation by minimal training, high reproducibility, rapid visual results, cost-effective, and easily quantified if equipped with an inexpensive portable, hand-carry detector. The major disadvantage of using LFIA was indicated by previous workers that only qualitative, and at best semi-quantitative results were obtained; nevertheless, we have repetitively proven in our hand that this modified LFIA can be reproducibly used for obtaining quantitative results. Other major disadvantage of using traditional LFIA was that results were variable among different batches of materials and limited by different antibodies used. This has been overcome in our modified LFIA that we have an ample supply of our home-made chicken antibody [9,24] and different alternatives of commercial rabbit antibodies [17,18,21,22,23]. To ensure continuous supply of antibodies, we are currently investing in the development of monoclonal antibodies in the near future. The potential limitation for using the METCAM/MUC18 as a diagnostic biomarker for predicting malignant potential of prostate cancer is that it is not prostate organ-specific, since it is also overly expressed in other cancers, such as angiosarcoma, breast cancer, gastric cancer, hepatocellular carcinoma, lung cancers, most melanoma, nasopharyngeal carcinoma type III, osteosarcomas, and pancreatic cancer; however, it is under expressed in some cancers, such as colorectal cancer, nasopharyngeal carcinoma type I, and ovarian cancer [17,26]. Nevertheless, it may be feasible to use the METCAM/MUC18 to at least complement the current biomarker PSA and furthermore, to combine it with other prostate cancer biomarkers for accurately predicting the malignant potential of prostate cancer [4,5,6,7,8,18].

The success of this assay has fulfilled the main purpose of this manuscript that is to have a strong proof of principle in using METCAM/MUC18 as a novel biomarker for predicting the prostate cancer at the early premalignant stage. Furthermore, it has encouraged us and investigators from other groups to have confidence in further validating this bio-marker for predicting the malignant potential of prostate cancer at the early premalignant PIN stage. We will continue expanding our survey cohort in Taiwanese males to further validating the use of METCAM/MUC18 as an alternative biomarker for the early diagnosis of prostate cancer. We will also use this method to explore the possibility of detecting the presence of METCAM/MUC18 antigen in urine samples [27]. If this turns out to be successful, it may be used to replace the current invasive method to obtain clinical specimens for an effective diagnosis of prostate cancer.

## 5. Conclusions

The rule of thumb for treating prostate cancer or any other cancers is that early detection of any cancer is always better. Because then the treatment can be performed at early stage when the cancer cells still are confined in the organ and less likely to metastasize outside of the organ and therefore, the curative rate is always very high, which is supported by numerous clinical evidences that prostate cancer at an early stage is entirely curable, but not the cancer at late stage. To meet the above purpose, first, this modified LFIA had been greatly improved from the traditional LFIA to become an accurate, simple, and rapid diagnostic method for comparing the concentrations of the human METCAM/MUC18 antigens in serums from normal human individuals with those from prostate cancer patients. Second, this simplified diagnostic assay can be used for accurate diagnosis of the malignant potential of prostate cancer at the pre-malignant (PIN) stage before becoming a frank malignant cancer (namely a metastatic cancer), which is different from the PSA test, suggesting that METCAM/MUC18 appears to have the advantage over the PSA test in early detection of the malignant potential of prostate cancer. Considering the ability of METCAM/MUC18 as a driver for the malignant progression of the cancer, after extensive follow-up checking of our patients in future, we also believe that METCAM/MUC18 has the high potential to be used for differentiating the indolent prostate cancers from aggressive ones in clinics.

## Figures and Tables

**Figure 1 diagnostics-11-00443-f001:**
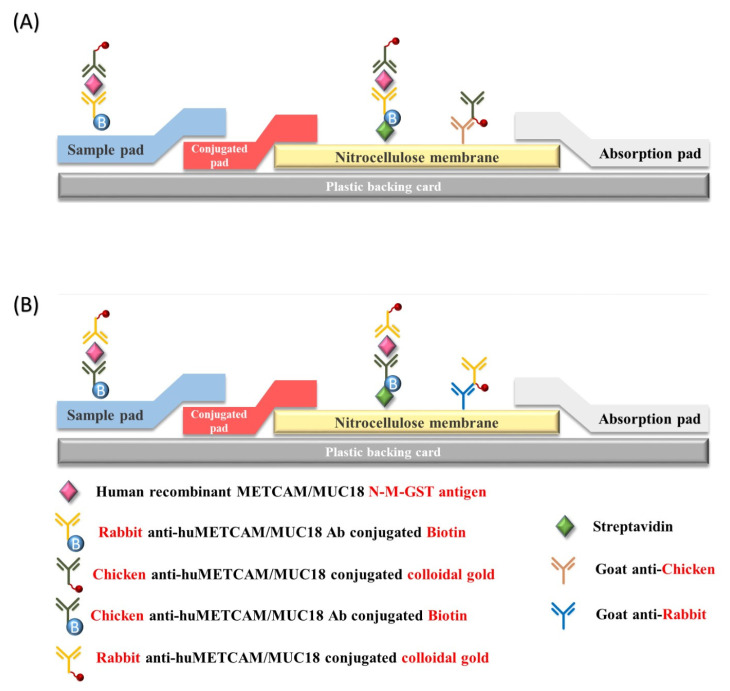
The set-up of modified Lateral Flow Immune Assay (LFIA). LFIA was set up by assembling of the sample pad, the conjugation pad, the nitrocellulose membrane, and the absorption pad on top of a plastic backing card. Streptavidin was printed on the T-line of and goat antibody against primary chicken antibody or rabbit antibody printed on the C-line of the nitrocellulose membrane. (**A**) shows the set-up of modified LFIA by using the first antibody combination (commercial biotinylated rabbit antibody EPP11278 and nano-gold conjugated home-made chicken antibody). (**B**) shows the set-up of modified LFIA by using the second antibody combination (biotinylated home-made chicken antibody and nano-gold conjugated commercial rabbit antibody EPP11278).

**Figure 2 diagnostics-11-00443-f002:**
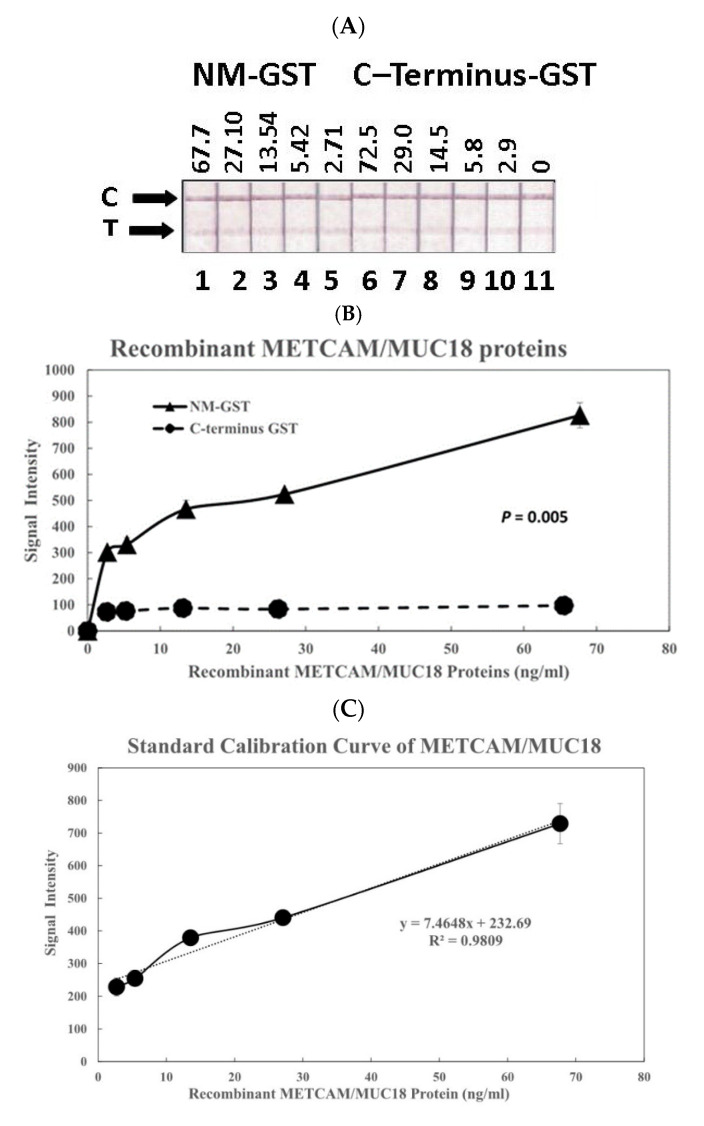
(**A**) Recombinant std protein image combo1, (**B**) Positive and negative controls combo1, (**C**) Calibration curve combo 1. Establish the calibration curve from recombinant huMETCAM/MUC18 proteins as determined by a modified LFIA method using the combination of biotinylated rabbit antibody (EPP11278) and nanogold-conjugated home-made chicken antibody. (**A**) shows the intensities of the two recombinant METCAM/MUC18 proteins, the positive control protein, NM-GST (N-terminus and the middle portions fused to GST), and the negative control protein, C-terminus-GST (the C-terminus portion fused to GST), at different concentrations. C stands for control lines and T for test lines. (**B**) shows the quantitative results of the intensities of the two recombinant METCAM/MUC18 proteins at different concentrations. (**C**) shows the standard calibration curve obtained by subtracting the intensity of the negative control protein from the positive control protein.

**Figure 3 diagnostics-11-00443-f003:**
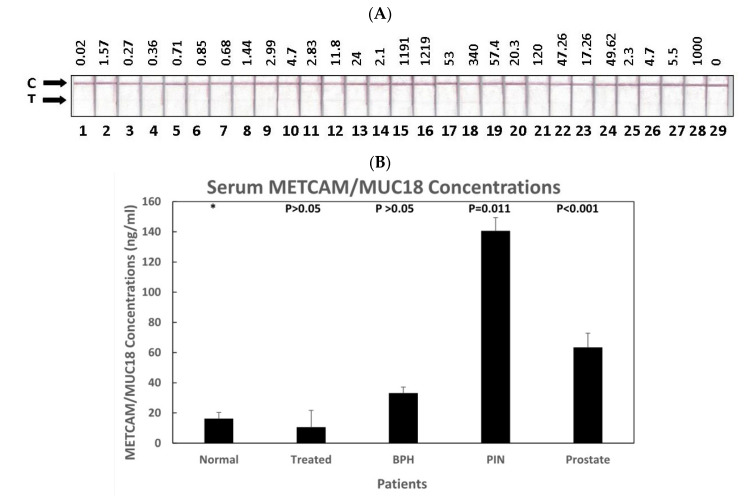
(**A**) Serum image for Combo1, (**B**) METCAM in patients Combo1, (**C**) METCAM in patients with Gleason score combo1. HuMETCAM/MUC18 concentrations in various human serum samples were determined by a modified LFIA method using the combination of biotinylated rabbit antibody (EPP11278) and nanogold-conjugated chicken antibody and its relation to the status of patients. (**A**) shows the intensities of various human serum samples on the test lines (T). The top numbers show the prostate-specific antigen (PSA) score for each patient. (**B**) shows the average METCAM/MUC18 concentrations with standard deviations in the serum samples from the treated prostate cancer patients, prostate cancer patients, patients with prostate premalignant prostatic intraepithelial neoplasia (PIN), or benign prostatic hyperplasia (BPH), were compared to those from normal individuals. The number on top of each bar was the *p* value when the data were compared to the normal after statistical analysis, as described in “Materials and Methods.” (**C**) shows the average METCAM/MUC18 concentrations with standard deviations in the serum samples from the treated prostate cancer patients, prostate cancer patients with different Gleason score, and patients with prostate premalignant PIN, or BPH, compared to those from normal individuals. The number on top of each bar was the *p* value when the data were compared to the normal after statistical analysis, as described in “Materials and Methods”.

**Figure 4 diagnostics-11-00443-f004:**
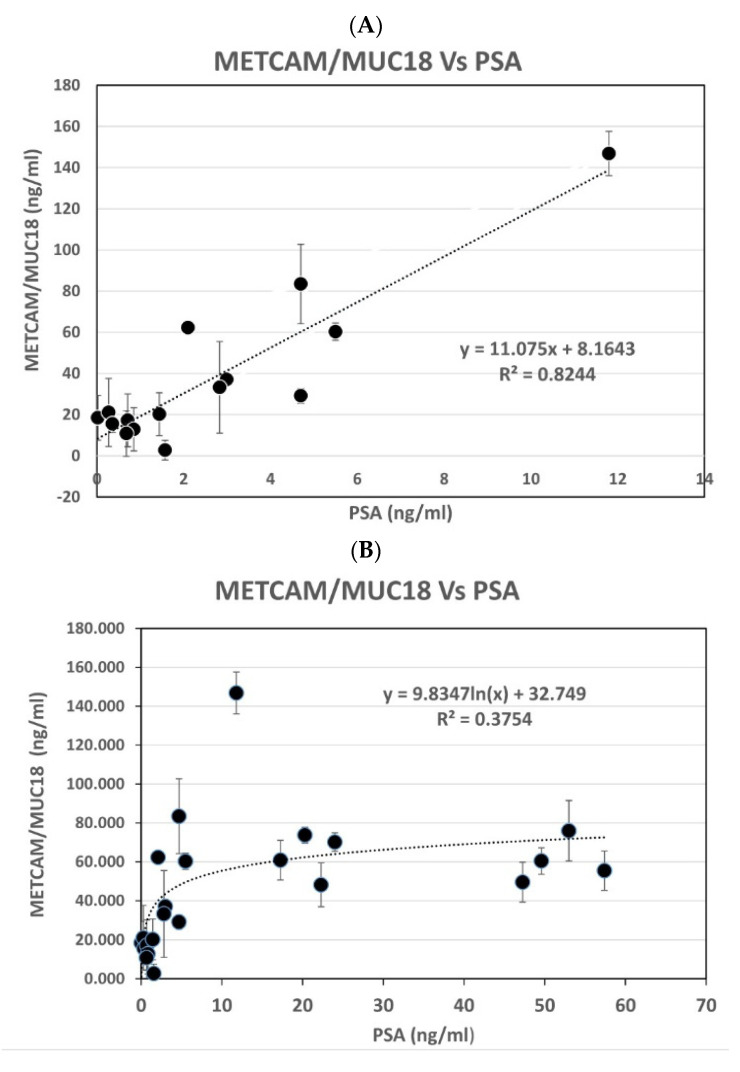
(**A**) METCAM vs. PSA (#1) combo1, (**B**). METCAM vs. PSA (#2) combo1, (**C**) METCAM vs. PSA (#3) combo1, (**D**) METCAM vs. PSA (#4) combo1. Relation of PSA to the huMETCAM/MUC18 concentrations determined by the modified LFIA method by using the combination of biotinylated rabbit antibody and nanogold-conjugated chicken antibody. The concentration of serum METCAM/MUC18 was compared to PSA in each person. (**A**) shows the results of serum METCAM/MUC18 concentrations versus PSA when PSA was less than 12 ng/mL, (**B**–**D**) show the results of serum METCAM/MUC18 concentrations versus PSA when PSA were less than 60 ng/mL, less than 350 ng/mL, and less than 1219 ng/mL, respectively.

**Figure 5 diagnostics-11-00443-f005:**
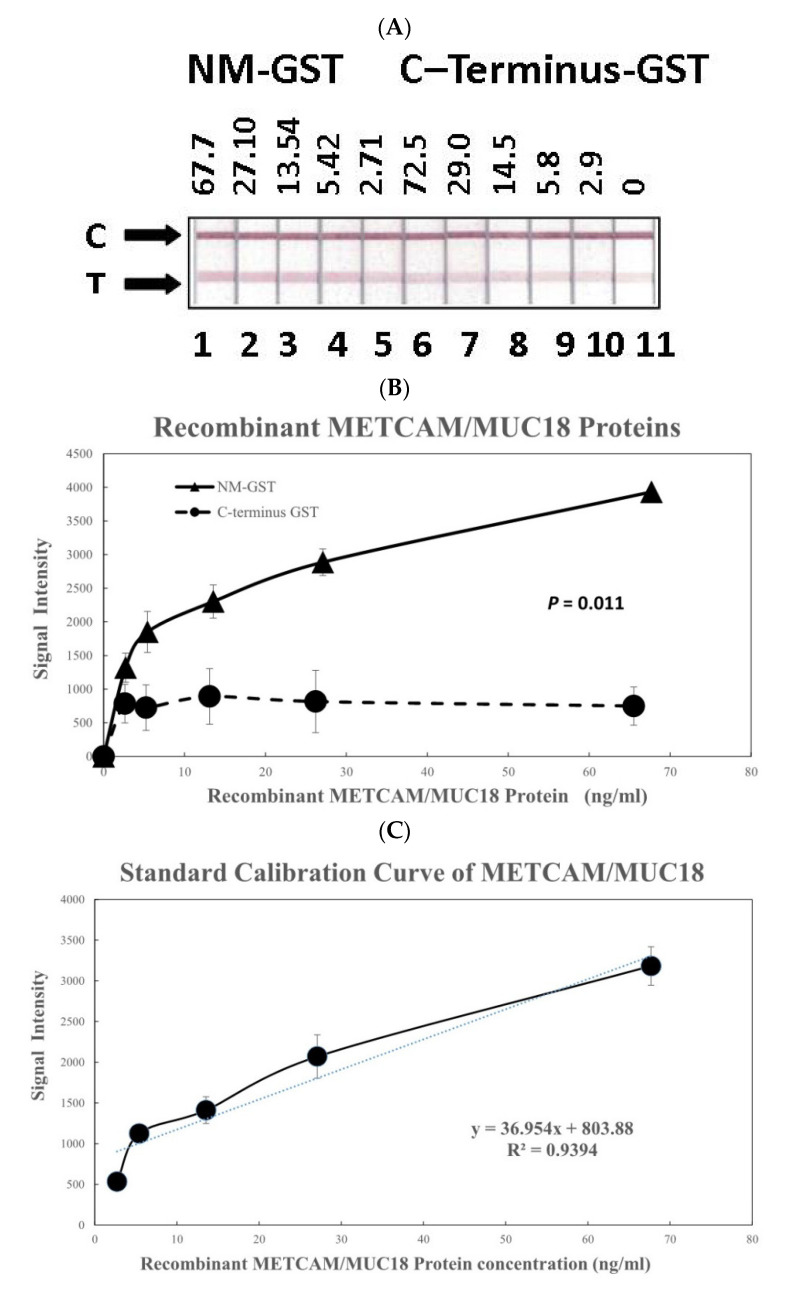
(**A**) Recombinant std protein image combo2, (**B**) Positive and negative controls combo2, (**C**) Calibration curve combo2. Establish the calibration curve from recombinant METCAM/MUC18 proteins as determined by a modified LFIA method by using home-made biotinylated chicken antibody and nanogold-conjugated rabbit antibody (EPP11278). (**A**) shows the intensities of the recombinant METCAM/MUC18 proteins, the positive control protein, NM-GST, and the negative control protein, C-terminus-GST, at different concentrations. C stands for control lines and T for test lines. (**B**) shows quantitative results of the intensities of the two recombinant METCAM/MUC18 proteins at different concentrations. (**C**) shows the standard calibration curve obtained by subtracting the intensity of the negative control protein from the positive control protein.

**Figure 6 diagnostics-11-00443-f006:**
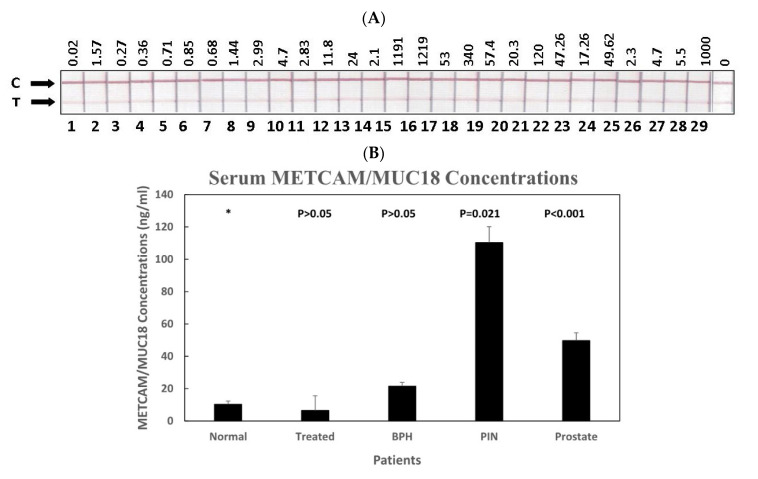
(**A**) Serum image for Combo2, (**B**) METCAM in patients Combo2. HuMETCAM/MUC18, (**C**) METCAM in patients with Gleason score combo2. HuMETCAM/MUC18 concentrations in various human serum samples were determined by a modified LFIA method using the combination of biotinylated chicken antibody and nanogold-conjugated rabbit antibody (EPP11278) and its relation to the status of patients. (**A**) shows the intensities of various human serum samples on the test lines (T). (**B**) shows the average METCAM/MUC18 concentrations with standard deviations in the serum samples from the treated prostate cancer patients, prostate cancer patients, patients with prostate premalignant PIN, or BPH, were compared to those from normal individuals. The number on top of each bar was the *p* value when the data were compared to the normal after statistical analysis, as described in “Materials and Methods.” (**C**) shows the average METCAM/MUC18 concentrations with standard deviations in the serum samples from the treated prostate cancer patients, prostate cancer patients with different Gleason score, and patients with prostate premalignant PIN, or BPH, compared to those from normal individuals. The number on top of each bar was the *p* value when the data were compared to the normal after statistical analysis, as described in “Materials and Methods”.

**Figure 7 diagnostics-11-00443-f007:**
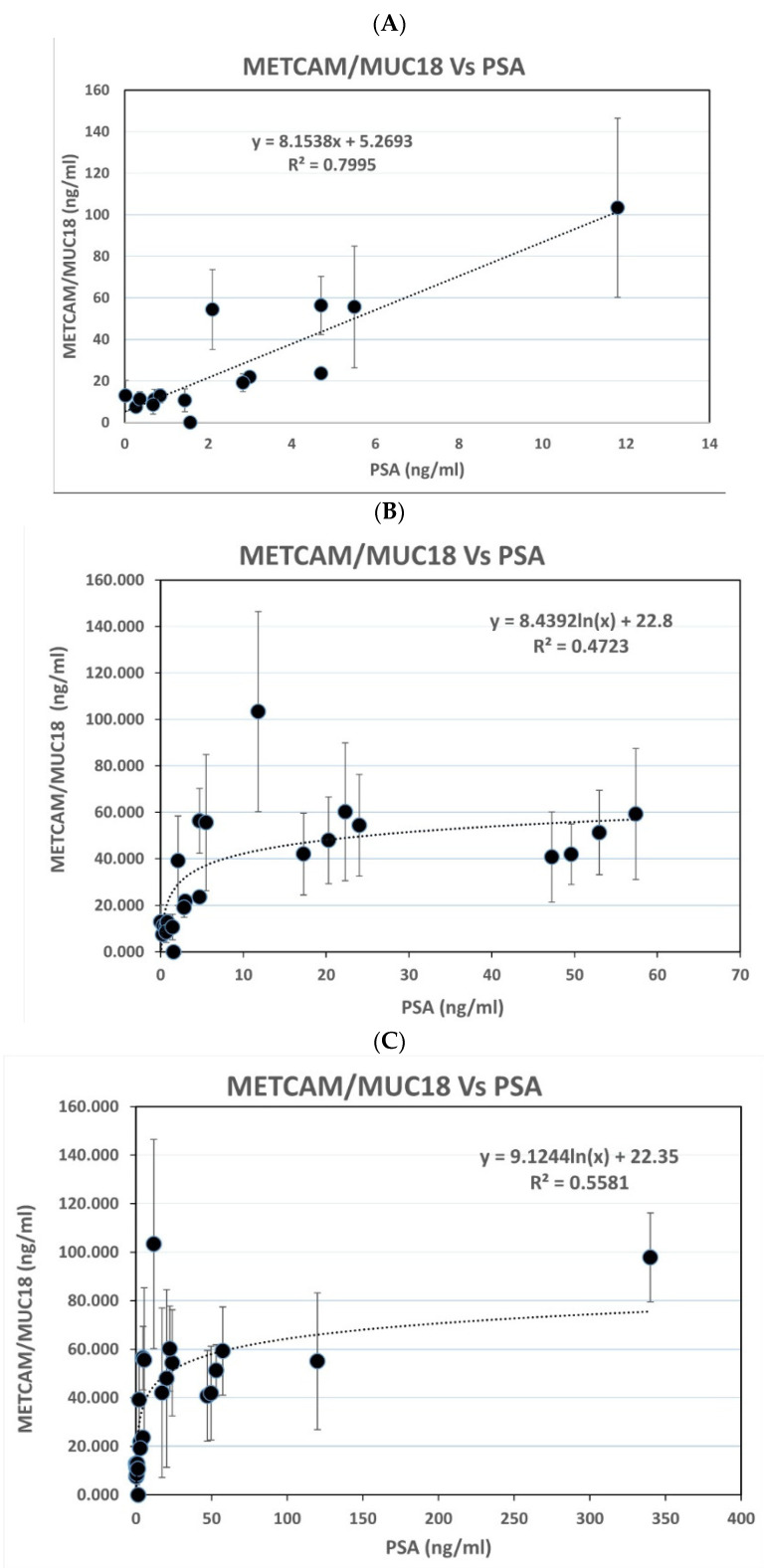
(**A**) METCAM vs. PSA (#1) combo2; (**B**). METCAM vs. PSA (#2) combo2, (**C**). METCAM vs. PSA (#3) comb2, (**D**). METCAM vs. PSA (#4) comb2. Relation of PSA to the huMETCAM/MUC18 concentrations determined by the modified LFIA method by using the combination of biotinylated chicken antibody and nanogold-conjugated rabbit antibody (EPP11278). The concentration of serum METCAM/MUC18 was compared to PSA in each person. (**A**) shows the results of serum METCAM/MUC18 concentrations versus PSA when PSA was less than 12 ng/mL, (**B**–**D**) show the results of serum METCAM/MUC18 concentrations versus PSA when PSA was less than 60 ng/mL, less than 350 ng/mL, and less than 1219 ng/mL, respectively.

**Figure 8 diagnostics-11-00443-f008:**
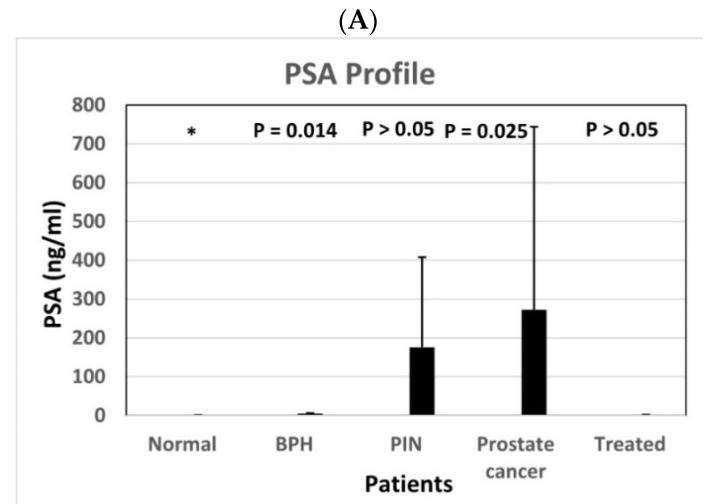
(**A**) PSA profile in patients, (**B**) PSA profile in patients with Gleason score, Serum PSA concentrations in various patients. (**A**) shows the average PSA concentrations with standard deviations in the serum samples from the treated prostate cancer patients, the patients with prostate cancer, the patients with premalignant PIN, or the patients with BPH in comparison to those from normal individuals. (**B**) shows that the average PSA concentrations with standard deviations in the serum samples from the treated prostate cancer patients, prostate cancer patients with different Gleason score, and patients with prostate premalignant PIN, or BPH, compared to those from normal individuals. The number on top of each bar was the *p* value when the data were compared to the normal after statistical analysis, as described in “Materials and Methods”.

**Table 1 diagnostics-11-00443-t001:** Epitopes recognized by various antibodies.

Antibodies	Size of METCAM/MUC18 (Recognized in WB)	Location of Epitopes (# Amino Acid)	Recombinant Protein of huMETCAM/MUC18 Recognized in WB *	Sources
Chicken Ab	113–150 kDa	212–374	M	Home-made
Biotinylated chicken Ab	113–150 kDa	212–374	M	Home-made
Rabbit Ab (MBS2529469 = EPP11278)	120 kDa	26–350	N	MyBiosource and Elabscience
Biotinylated rabbit Ab (EPP11278)	120 kDa	26–350	N	Elabscience

* Recombinant proteins of human METCAM/MUC18: N terminus: aa# 1–212, M portion: aa# 212–374, NM portion: aa# 1–374, C-terminus: aa# 375–646.

**Table 2 diagnostics-11-00443-t002:** Patient (male) Characteristics (N = 30).

Characteristics	Age Range (Years)	Age Median (Years)	No of Cases	Percentage (%)
Normal	28–75	52	8	27
BPH	68–75	72	4	13.3
PIN	63–87	75	2	6.7
Prostate carcinoma (total)	50–93	72	14	47
Radiotherapy	67–73	70	2	6.7
Pathological grades of Prostate carcinoma	50–93	72	14	47
Gleason score 3 + 3	93	93	1	3.3
Gleason score 3 + 4 or 4 + 3	63–85	74	6	20
Gleason score 4 + 4	50–75	63	4	13.3
Gleason score 4 + 5 or 5 + 4	69–86	78	2	6.7
Gleason score 5 + 5	83	83	1	3.3

## Data Availability

All the data utilized in this study are available upon request to the corresponding author.

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
