# Peer review of "Validating METCAM/MUC18 as a Novel Biomarker to Predict the Malignant Potential of Prostate Cancer at an Early Stage by Using a Modified Gold Nanoparticles-Based Lateral Flow Immunoassay"

_diagnostics, 2021, doi:10.3390/diagnostics11030443_

Round 1
Reviewer 1 Report
This research addresses the shortcomings of the classical PSA test routinely used for prostate cancer diagnosis. The data provides clear evidence that this assay has potential to be a valuable tool in the battery of tests to used to confirm prostate cancer, notably at at an early stage. The assay is fully described and merits publication, and should be of interest to the oncology and pathology communities.
Figure one (illustration of the assay set-up) is a helpful inclusion and the paper distinguishes this new development from earlier work by this group; however, I draw attention to the error under the Figure (Fig. 1.) namely, there is a redundant entry "Fig. 8"? which requires attention.
Author Response
Responses to the comments of the Reviewer #1:
Thank you for your positive review for the overall content of the manuscript.
The spelling checks were performed again.
I am sorry for my oversight during format changes for Fig. 1. Thank you for pointing it out for my attention. I did correct the Fig. 8 to Fig. 1 under the figure. Thank you very much.

Reviewer 2 Report
The paper is in an excellent form
Author Response
Response to the comment of the Reviewer #2:
Thank you for your positive review of the overall content of the manuscript. I appreciate your confidence in it and laudable honesty.

Reviewer 3 Report
Review
The authors presented a study on validating of novel biomarker METCAM/MUC18 of prostate cancer by a modified Lateral Flow Immune Assay (LFIA). It was shown that METCAM/MUC18 level in patients with PIN was consistently higher than the prostate cancer patients and significantly higher in patients with cancer and PIN than in men without prostate pathology or with BPH. Further studies of the biomarker are of particular interest for screening purposes. However, there are some issues that should be addressed
1) In the article the serum METCAM/MUC18 concentrations in the patients with PIN were higher than in patients with prostate cancer [1]. The authors suggested addressing why the new assay detects PIN better than PCa.
2) Only 30 blood samples of patients were analyzed; only 2 of them were from patients with PIN and 14 with BPH. This amount of data seems to be limited for drawing adequate conclusions.
3) Authors suggested giving a detailed explanation of why the ability of new assay to detect PIN is of clinical importance.
[1] Pong, Y.H.; Su, Y.R.; Lo, H.W.; Ho, C.K.; Chu, C.T.; Chen-Yang, Y.W.; Tsai, V.F.S.; Wu, J.C.; Wu, G.J. METCAM/MUC18 is a new early diagnostic biomarker for the malignant potential of prostate cancer: validation with Western blot method, enzyme-linked immunosorbent assay and lateral flow immunoassay. Cancer Biomarkers 2020, 27, 377-387.
Author Response
Responses to the comments of the Reviewer #3.
The English language has been thoroughly checked again.
Thank you for your comments and especially compared the manuscript to my previous paper (Cancer Biomarkers 2020).
Your comments for the overall contents of the manuscript indicated that research design, results, and conclusions must be improved, even though the other two reviewers had a total understanding of and complete confidence in the contents of the manuscript and thus had very positive comments in all sections. Nevertheless, your comments remind me that I should improve further in these sections, especially the following three points for those who did not have enough background in cancer biology.
- If you have paid a close attention to the details of the previous paper (Pong et al., Cancer Biomarkers 2020, 27, 377-387), you would find the results of the PIN serum were somewhat statistically significant (p = 0.1), but not good enough for us to be convincing and conclusive because the data were obtained by using only the Western blot method for analyses, in which the data had a much larger standard deviation. Furthermore, the results were not reproduced and confirmed by using the other two more sensitive methods (ELISA and LFIA) for analyses, in which the data had a smaller standard deviation, because unfortunately the two graduate students did not include the PIN serum samples for their tests. In the current manuscript, two reasons, in which the new assay detects PIN better than PCa, are (a) that we were lucky to have sera from two PIN patients for the analyses and the data were statistically significant (p = 0.011-0.021) and very convincing and conclusive to us, and (b) furthermore, the modified LFIA assay was improved from a traditional LFIA by using the high affinity biotin-streptavidin interaction in the assay to increase sensitivity and specificity, thus the data were more reproducible and reliable with even a smaller standard deviation (this point has already been described in the original version, as shown in red font in this revised version).
- I agreed with you that if we could obtain more serum samples, especially from the PIN patients, it would be better. However, in reality it is more difficult to obtain many PIN serum samples than more numerous prostate cancer serum samples, because most of the prostate cancer patients came to clinics were in later stages, especially in Taiwan (Chen et al., Int J Cancer 2013, 132, 1927-1932). Thus, we were lucky to obtain two PIN sera for the analysis. Regardless, 30 serum samples (8 from normal individuals, 2 from PIN, 4 from BPH and 14 from prostate cancer patients and 2 from treated patients, not as what you mentioned that 2 of them were from patients with PIN and 14 with BPH) were shown to be acceptable and more than enough to draw significant conclusion in the numerous published works that carried out the phase I (several patients) and phase II (25-100 patients) tests in clinical trials. Furthermore, this amount of data from 30 serum samples seems to be sufficient to draw adequate conclusions because the data were shown to be statistically significant (p = 0.011-0.021 for PIN versus normal and p < 0.001 for PCa versus normal). Most importantly, we have fulfilled the main purpose of this manuscript that is to have a strong proof of principle in using METCAM/MUC18 as a novel biomarker for predicting the prostate cancer at the early premalignant stage; thus, we and investigators from other groups are encouraged to have confidence in further validating this bio-marker for predicting the malignant potential of prostate cancer at the early premalignant PIN stage. This point is re-emphasized in the revised version (as shown in red font in “Discussion”).
- The rule of thumb for treating prostate cancer or any other cancers is that early detection of any cancer is always better. Because then the treatment can be performed at early stage when the cancer cells still are confined in the organ and less likely to metastasize outside of the organ and therefore, the curative rate is always very high, which is supported by numerous clinical evidences that prostate cancer at an early stage is entirely curable, but not the cancer at late stage. We should strongly re-emphasize this point in the revised conclusion (as shown with red font in “Conclusions”).

Round 2
Reviewer 3 Report
The authors addressed all the previously raised concerns, making the paper suitable for publication in the Diagnostics.